https://doi.org/10.1038/s41467-019-11976-2　　**OPEN**

# A general strategy for diversifying complex natural products to polycyclic scaffolds with medium-sized rings

Changgui Zhao [1], Zhengqing Ye [1], Zhi-xiong Ma [1], Scott A. Wildman [2], Stephanie A. Blaszczyk [3], Lihong Hu[4], Ilia A. Guizei [3] & Weiping Tang [1,3]

The interrogation of complex biological pathways demands diverse small molecule tool compounds, which can often lead to important therapeutics for the treatment of human diseases. Since natural products are the most valuable source for the discovery of therapeutics, the derivatization of natural products has been extensively investigated to generate molecules for biological screenings. However, most previous approaches only modified a limited number of functional groups, which resulted in a limited number of skeleta. Here we show a general strategy for the preparation of a library of complex small molecules by combining state-of-the-art chemistry – the site-selective oxidation of C-H bonds - with reactions that expand rigid, small rings in polycyclic steroids to medium-sized rings. This library occupies a unique chemical space compared to selected diverse reference compounds. The diversification strategy developed herein for steroids can also be expanded to other types of natural products.

---

[1] School of Pharmacy, University of Wisconsin-Madison, Madison, WI 53705, USA. [2] Carbone Cancer Center, School of Medicine and Public Health, University of Wisconsin, Madison, WI 53705, USA. [3] Department of Chemistry, University of Wisconsin-Madison, Madison, WI 53706, USA. [4] Jiangsu Key Laboratory for Functional Substance of Chinese Medicine, School of Pharmacy, Nanjing University of Chinese Medicine, 210023 Nanjing, P. R. China. Correspondence and requests for materials should be addressed to W.T. (email: wtang@pharmacy.wisc.edu)

Medium-sized rings are largely under-represented in current screening libraries for drug development, and not unexpectedly, they were rarely found in the top 200 brand name and top 200 generic pharmaceuticals[1]. While drug candidates may not enter the commercial market for many reasons, synthetic accessibility undoubtedly affects drug discovery and can bias subsequent drug development. Using state-of-the-art chemistry, we have developed a library of polycyclic medium-sized ring compounds that helps fill the void in an underexplored chemical space.

Natural products have an inherent diversity that is unparalleled by current synthetic libraries and their success rate as drug leads is unrivaled[2]. As such, they have been regarded as one of the most valuable sources for library generation[3] and lead compounds in drug development[4–9]. Natural products often contain privileged scaffolds, structural complexity, and abundant stereochemistry throughout the sp3-hybridized carbon atom skeleton[6,7]. The diversification of natural products has the potential to generate structurally complex small molecules with varied skeleta and ample stereochemical handles[7]. Synthetic strategies such as diversity-oriented synthesis (DOS)[10–12], biological-oriented synthesis (BiOS)[13], and function-oriented synthesis (FOS)[14] have been successfully executed to provide efficient access to many structurally complex and functionally diverse small molecules[15–20]. However, most of the synthetic strategies published to date suffer from a significant limitation—they rely on the presence and reactivity of specific functional groups (FGs) that are only present in certain classes of natural products.

Conversely, C–H bonds are ubiquitous and exist in almost every organic compound. For this reason, C–H functionalization has become one of the major focuses in organic chemistry. C–H functionalization chemistry has been applied to the synthesis of a number of natural products and pharmaceutical reagents[21], including target-oriented synthesis and last-stage functionalization of natural products[22]. We envisioned that the diversification of natural product skeleta via C–H functionalization would offer a general strategy for the synthesis of many structurally complex and functionally diverse natural product-like small molecules, while simultaneously preparing compounds in an underexplored chemical space.

Following C–H functionalization, we plan to use the resulting functional group handles for ring expansion reactions to prepare medium-sized rings, which are core components in bioactive natural products and medicinally relevant compounds[23]. Due to unfavorable transannular interactions and entropic effects[24], the synthesis of medium-sized rings has been a longstanding challenge in organic chemistry. Even now, general strategies for their formation are limited[25,26]. Medium-sized rings aren't subject to the rigidity of small rings, yet they lack the flexibility of linear compounds or macrocycles. Furthermore, their reactivity can be difficult to predict owing to the presence of multiple low energy conformations[27]. These unique properties make them attractive yet challenging targets.

To reach these targets we employed nature's most robust molecular family—terpenes, but more specifically steroids. Of all natural product families, the terpenes exhibit unparalleled chemical and structural diversity[28,29]. Their biosynthesis involves two major phases: cyclization of hydrocarbons by cyclase and oxidation by cytochrome P450 (CYP) enzymes[30]. The chemistry underpinning their biosynthesis, particularly the oxidation component, inspired Baran's group to complete several efficient total syntheses of complex natural products by preparing the hydrocarbon skeleta with low oxidation stage followed by selective C–H oxidation[31–36]. Steroids are perhaps the most privileged structures in the history of drug development. In fact, more than 100 different steroid molecules have been approved by the U.S. Food and Drug Administration (FDA) as therapeutics for a wide range of symptoms and diseases, such as inflammation, pain, cancers, and bacterial infections[37]. Not surprisingly, a variety of methods have been developed for the modification of steroids. However, most of these methods focused on simple modifications repeatedly using the same core skeleta and FGs[38,39]. Examples of significant diversification are rare[37,40].

Inspired by nature's biosynthesis pathways and Baran's two-phase strategy for the total synthesis of terpenes[31–36], we envisioned that oxidation of the most abundant C–H bonds in polycyclic natural products to C–O bonds, followed by ring expansion, may offer a general and efficient strategy for the diversification of natural products and yield a collection of compounds to probe an underexplored chemical space—polycyclic compounds with medium-sized rings (Fig. 1). To probe the feasibility of this two-phase diversification strategy, we investigated the viability of synthesizing medium-sized rings via ring expansion reactions with substrates featuring native C–O bonds, as in hydroxyl or ketone groups. Success in this endeavor then led us to use a sequential C–H oxidation/ring expansion strategy to diversify polycyclic natural products.

Our strategy is a hybrid of target and diversity-oriented synthesis. We targeted the formation of medium-sized rings yet wanted to diversify our products as well. Innovative strategies, such as these, are crucial to making compound libraries that represent underexplored chemical space. We herein report our diversification strategy for the synthesis of natural product-like, polycyclic scaffolds containing medium-sized rings (7–11 membered) using ring expansion reactions and C–H functionalization.

## Results

**Ring expansion based on native C–O bonds.** Our initial studies on the ring expansion strategy used four steroid natural products —dehydroepiandrosterone (DHEA), cholesterol, isosteviol, and estrone—all of which have been reported to exhibit broad pharmacological activities. As shown in Fig. 2, the A-rings of **1a** and **1b** were expanded to [5.3.0] fused bicycles **3a** and **3b** through an intramolecular Schmidt reaction. Reduction of the ketone in **3a** provided alcohol **4a** as a single isomer, whose relative stereochemistry was confirmed by X-ray crystallography. The stereochemistry of **4b** was assigned by analogy. The β-keto esters **2a**, **2b**, and **2c** could undergo a formal [2 + 2] cycloaddition reaction with dimethyl acetylenedicarboxylate (DMAD) leading to ring-expanded compounds **5a-c** after fragmentation of the cyclobutene moiety to generate the corresponding two-carbon ring-enlargement compound. The esters **5a** and **5c** were then further elaborated to give [6.3.0] fused bicyclic anhydrides **6a** and **6c**. Fused bicyclic anhydride moieties present in natural products have been reported to exhibit a diverse range of bioactivity[41], and the anhydrides can also be easily derivatized, as discussed later (Fig. 2a, Supplementary Fig. 5). The bridged five-membered ring of isosteviol **1d** could be expanded to six-membered β-keto ester **7** by treating with BF3•Et2O and ethyl diazoacetate. Ester **7** could also undergo ring expansion upon reaction with DMAD and followed by AcOH/HCl to give [6.3.0] fused bicyclic anhydride **8**. The six-membered ring in **7** could also be converted to lactam **9** by a Beckmann rearrangement (Fig. 2b, Supplementary Fig. 6). The D-ring of **13** could be expanded to benzannulated medium-sized ring **14** with high diastereoselectivity (d.r. = 18:1) via ring expansion/aryne insertion chemistry. The relative stereochemistry of **14** was confirmed by NOE experiments. Interestingly, the major stereoisomer **14** could be isomerized to the minor stereoisomer **15** using TBAF in THF. The expansion of the D-ring in estrone (Fig. 2c, Supplementary Fig. 7) and DHEA (Fig. 2d, Supplementary Fig. 7) could be realized by an acylation/ring

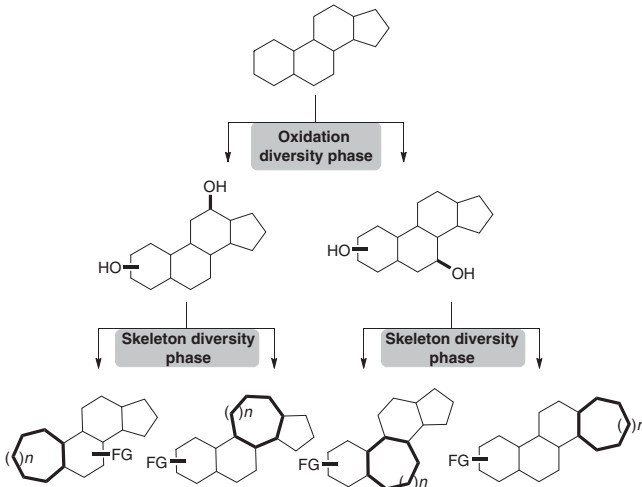

**Fig. 1** Two Phases of Diversification. Starting from a polycyclic scaffold, we are able to oxidize then functionalize various natural products to produce modified scaffolds

expansion sequence leading to nine-membered rings bearing a β-keto ester motif within polycyclic compounds **12** and **17**[42]. Furthermore, the newly formed medium-sized ring **17** could then undergo an additional Schmidt reaction leading to **18** (Fig. 2d, Supplementary Fig. 7). The D-ring of DHEA could also be expanded to other medium-membered ring scaffolds through a similar two-carbon ring expansion strategy (Fig. 2d, Supplementary Fig. 7). The transformations shown in Fig. 2 indicate both the feasibility and generality of using native C–O bonds for ring expansion to form medium-sized rings in natural product diversification.

**Sequential C–H oxidation/ring expansion**. C–H functionalization reactions, especially C–H oxidation, have recently emerged as a powerful strategy for the derivatization of complex organic compounds. After determining that polycyclic steroids containing native C–O bonds could be expanded to contain medium-sized rings, we significantly diversified natural products by applying a two-phase strategy—installation of C–O bonds via C–H oxidation followed by ring expansion. Among the C–H oxidation methods, electrochemical oxidation generates less toxic waste and is compatible with a wide range of functional groups, making it an attractive complementary strategy to traditional chemical reagents[43,44]. Employing the electrochemical allylic C–H oxidation method developed by Baran[45,46], this procedure, in conjunction with a Beckmann rearrangement, allowed us to synthesize several seven-membered lactams (Fig. 3a, b, Supplementary Fig. 8, 9). Using a site-selective copper-mediated C–H oxidation[47] and Beckmann rearrangement, we expanded the C-ring of estrone and DHEA (Fig. 3c, Supplementary Fig. 10). Lastly, we used a chromium-mediated benzylic C–H oxidation[48] of **1j** to yield ketone **28**, which provided lactam **29** after a Beckmann rearrangement (Fig. 3d, Supplementary Fig. 11).

**Diversification of more complex natural products**. We next examined the scope of the C–H oxidation/ring expansion strategy by applying it to the diversification of two significantly more complex natural products—picfeltarraegenin **30** and kirenol **36**. Protection and reduction with NaBH₄ converted natural product **30** to **32**. From here, the electrochemical allylic C–H oxidation/ring expansion strategy was successfully applied to the synthesis of seven-membered lactam-containing compound **35**, whose relative stereochemistry was confirmed by NOE experiments

(Fig. 4a, Supplementary Fig. 12). Protection of **36** by cyclic carbonate and MOM ethers converted natural product **36** to **37**. Allylic C–H oxidation of **37**, followed by oxidation of the OH group, afforded enone **38**. Finally, a Beckmann rearrangement yielded lactam **39**, whose relative stereochemistry was confirmed by NOE experiments (Fig. 4b, Supplementary Fig. 12).

**Further diversification of medium-sized rings**. Previously synthesized products containing medium-sized rings offered multiple opportunities for further functionalization. To demonstrate the feasibility of our C–H oxidation/ring expansion strategy for the preparation of compound libraries, we further derivatized some of the functional groups to yield a collection of 150 compounds (Fig. 5). For example, hydroxyl groups were derivatized to carbamates, β-keto esters underwent alkylation, anhydrides were converted to imides, and ring-opening reactions were performed on some of the medium-sized ring. Removal of the TBS protecting group provided the corresponding alcohols, which were then converted into a library of carbamates by treatment with CDI and a range of amines. (Fig. 5a, Supplementary Fig. 13). Alkylation of the β-keto ester scaffold afforded a single stereoisomer, which could also be converted to libraries of carbamates **45** and triazoles **46** (Fig. 5b, Supplementary Fig. 14). By reacting the anhydride of **6a** and **8** with diverse primary amines, libraries of imides **47** and **48** were prepared (Fig. 5c, d Supplementary Fig. 15). Ring-cleavage of **5b** and **5c** provided **49**, which possesses three orthogonal FGs that can be further modified to larger collections of compounds. For example, direct reaction of the anhydride with amines resulted in the formation of imides **50**. Subsequently, the carboxylic acid moiety was easily converted to libraries of amides **51** and **52**. The alcohol moieties of libraries of **52** could react with CDI and amines to produce libraries of carbamates **54**. The carboxylic acid and alcohol FGs in library **50** could react with CDI, followed by treatment with amines to yield library **53** (Fig. 5e, Supplementary Fig. 16).

All of the compounds we prepared are natural product-like. They contain structural features often found in natural products, such as polycyclic ring systems, sp³ hybridized carbons, and abundant stereogenic centers, while also being densely populated with FGs capable of hydrogen bonding. Since medium-sized rings can adopt multiple low energy conformations, these compounds may exhibit multiple modes of binding in biological contexts. Most remarkably, a large number of scaffolds with diverse structural features can be accessed using a relatively small set of C–H oxidation and ring expansion reactions.

**Chemoinformatic analysis**. We used principal component analysis (PCA)[49] to compare our compound library with approved therapeutics and natural products following the protocols employed by Tan[50]. We selected 30 representative compounds from our polycyclic, medium-sized ring library (Supplementary Fig. 3) and compared them with 40 top-selling small-molecule drugs in 2016 (without including the top-selling steroid drugs)[51], 25 top-selling small-molecule steroid drugs in 2016[51], a set of 25 steroid and terpenoid natural products with diverse oxidative stage and skeleta[28,29,38], and a set of 25 diverse natural products with a medium-sized ring[16,25,41,50]. The 40 top-selling small-molecule drugs in 2016 and the 25 top-selling small-molecule steroid drugs in 2016 are subsets of the 200 top-selling drugs compiled and produced by Njardarson's group[51]. The details of selection criteria for the natural products are described in the Supplementary Information (page S93). The 20 physicochemical parameters, such as molecular mass, slogP, hydrogen-bond donors/acceptors, etc. (Supplementary Table 1) of the compounds were analyzed using the molecular operating

**Fig. 2** Ring expansion of polycyclic natural products based on native C–O bonds. **a** A-ring expansion of dehydroepiandrosterone and cholesterol. **b** D-ring expansion of isosteviol. **c** D-ring expansion of estrone. **d** D-ring expansion of dehydroepiandrosterone. (i) (a) TPAP, NMO, 4 Å MS, CH₂Cl₂, (b) LDA, THF, then CNCO₂Et, −78 °C; (ii) (a) NaH, HMPA, THF, rt, then 1-Chloro-3-iodopropane, rt, (b) NaN₃, DMF, 80 °C, (c) CF₃COOH, rt; (iii) NaBH₄, MeOH, −78 °C; (iv) NaH, toluene, then DMAD, rt; (v) HCl, AcOH, 120 °C; (vi) (a) Me₂SO₄, LiOH, THF, 65 °C, (b) Ethyl diazoacetate, BF₃•Et₂O, Et₂O/CH₂Cl₂, rt; (vii) (a) LiCl, DMSO, H₂O, 120 °C, (b) NH₂OH•HCl, KOAc, 70 °C, (c) TsCl, DMAP, Py, 60 °C; (viii) Dibal-H, CH₂Cl₂, −78 °C; (ix) (a) NaOH, Me₂SO₄, acetone, 60 °C, (b) LDA, THF, −78 °C, then CNCO₂Et; (x) (a) MgCl₂, Py, NHCbz(CH₂)₂COCl, CH₂Cl₂, (b) Pd/C, H₂, EtOAc, rt; (xi) (a) TBSCl, imidazole, CH₂Cl₂, rt, (b) LDA, THF, then CNCO₂Et, −78 °C; (xii) 2-(Trimethylsilyl)phenyl trifluoromethanesulfonate, CsF, MeCN, 80 °C, 18:1 d.r.; (xiii) TBAF, THF, rt; (xiv) (a) NaH, THF, methyl phenylpropiolate, 65 °C, (b) TsOH, THF, rt. Full details are in the Supplementary Figures 5–7

environment (MOE) software. The first three principal components account for 72% of the dataset variance (Supplementary Table 3) and are represented in 2D plots (Fig. 6a–c, for expanded version see Supplementary Fig. 4a-c).

As shown in Fig. 6a–c, the parameters that have the greatest contributions to PC1 are H-Bond acceptors, number of rotatable bonds, polar surface area (tPSA) and molecular weight, which shift molecules right along the x axis in the plots of PC1/PC2 and

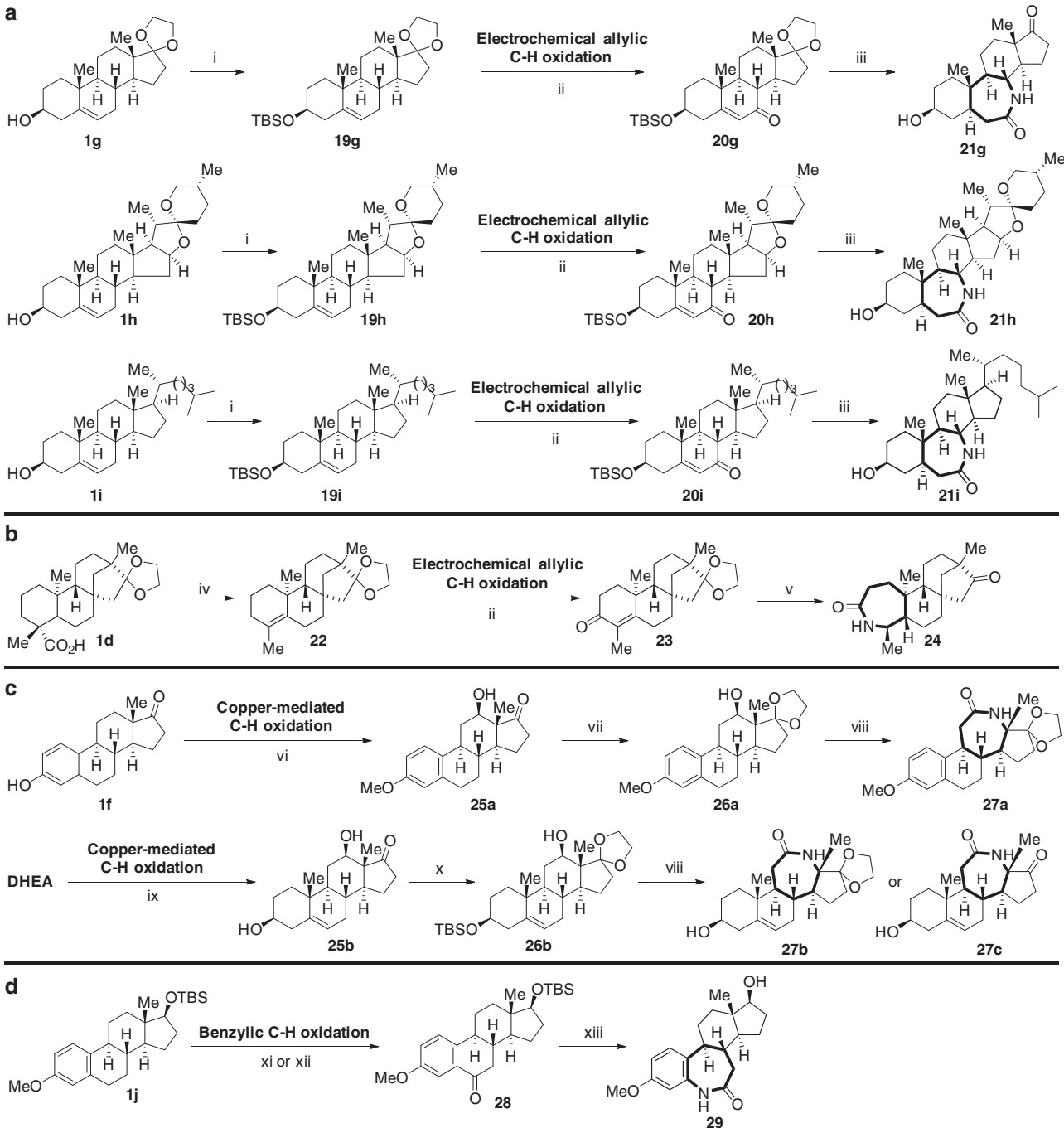

**Fig. 3** Diversification of polycyclic natural products by sequential C–H oxidation and ring expansion. **a** Electrochemical C–H oxidation/B-ring expansion of DHEA, cholesterol and Diosgenin. **b** Electrochemical C–H oxidation/A-ring expansion of isosteviol. **c** Copper-mediated C–H oxidation/C-ring expansion of DHEA and estrone. **d** Benzylic C–H oxidation/C-ring expansion of estrone. (i) TBSCl, imidazole, CH$_2$Cl$_2$, rt; (ii) LiClO$_4$, Py, $t$-BuOOH, Cl$_4$NHPI, acetone, rt; (iii) (a) Pd/C, H$_2$, EtOAc, rt, (b) NH$_2$O•HCl, KOAc, 70 °C, (c) TsCl, DMAP, Py, 60 °C; (d) TsOH, THF/H$_2$O or Pd/C, MeOH, rt; (iv) (a) Pb(OAc)$_4$, Cu (OAc)$_2$, Py, toluene, 90 °C, (b) I$_2$, toluene, 120 °C (c) glycol, toluene, TsOH, 120 °C; (v) (a) PtO$_2$, H$_2$, rt, (b) NaOMe, MeOH, rt, (c) NH$_2$OH•HCl, KOAc, 70 °C, (d) TsCl, DMAP, Py, 60 °C; (vi) (a) Me$_2$SO$_4$, K$_2$CO$_3$, acetone, (b) TsOH, toluene, (4-methylpyridin-2-yl)methanamine, 120 °C, (c) [Cu (MeCN)$_4$PF$_6$], O$_2$, (+)-sodium-(L)-ascorbate, acetone/MeOH, 50 °C, 6 h, then, Na$_4$EDTA, rt; (vii) glycol, toluene, TsOH, 120 °C; (viii) (a) (COCl)$_2$, DMSO, Et$_3$N, CH$_2$Cl$_2$, −78 °C (b) NH$_2$OH•HCl, KOAc, 70 °C, (c) TsCl, DMAP, Py, 60 °C, (d) TsOH, THF/H$_2$O, rt, for **27c**, 60 °C, 5 days; (ix) (a) TsOH, toluene, (4-methylpyridin-2-yl)methanamine, 120 °C, (b) [Cu(MeCN)$_4$PF$_6$], O$_2$, (+)-sodium-(L)-ascorbate, acetone/MeOH, 50 °C, 6 h, then, Na$_4$EDTA, rt; (x) (a) glycol, toluene, TsOH, 120 °C, (b) TBSCl, imidazole, CH$_2$Cl$_2$, rt; (xi) Cr(CO)$_6$, $t$BuOOH, MeCN, 70 °C; (xii) (a) K$^t$OBu, LDA, (MeO)$_3$B, THF, (b) H$_2$O$_2$, NaOH, (c) (COCl)$_2$, DMSO, Et$_3$N, CH$_2$Cl$_2$; (xiii) (a) NH$_2$OH•HCl, KOAc, EtOH, (b) TsCl, DMAP, Py, (c) CF$_3$COOH. Full details are in the Supplementary Figures 8–11

**Fig. 4** Application of the two-phase C–H oxidation/ring expansion strategy to picfeltarraegenin and kirenol. **a** Electrochemical allylic C–H oxidation/ring expansion of picfeltarraegenin. **b** Allylic C–H oxidation/ring expansion of kirenol. (i) MOMCl, DIPEA, DMAP, CH$_2$Cl$_2$, rt; (ii) (a) NaBH$_4$, EtOH, rt, (b) MOMCl, DIPEA, DMAP, CH$_2$Cl$_2$, 50 °C; (iii) LiClO$_4$, Py, $t$-BuOOH, Cl$_4$NHPI, acetone, rt. (iv) Pd/C, H$_2$, MeOH, rt; (v) NH$_2$OH•HCl, KOAc, (c) TsCl, DMAP, Py, 60 °C; (vi) (a) CDI, toluene, 90 °C, (b) MOMCl, DIPEA, DMAP, CH$_2$Cl$_2$, rt; (vii) (a) SeO$_2$, dioxane, (b) (COCl)$_2$, DMSO, Et$_3$N, CH$_2$Cl$_2$; (viii) (a) Pd/C, H$_2$, EtOAc, (b) NH$_2$OH•HCl, KOAc, 70 °C, (c) TsCl, DMAP, Py, 60 °C. Full details are in Supplementary Figure 12

PC1/PC3. The most significant contributors to PC2 are number of chiral centers, number of rings and chiral/weight. An increase in stereochemical complexity significantly shifts molecules downward in the plots of PC1/PC2 and PC2/PC3. Finally, the descriptors with largest loading on PC3 are rings, SLogP, aArRing (number of atoms in aromatic rings), and Oprea Ring Count[52]. These parameters have the effect of moving the molecules to the negative direction along the PC3 axis in the plots of PC1/PC3 and PC2/PC3, and the number of O atoms, which shifts compound in a positive direction in these plots.

As shown in Fig. 6a–c, there are significant overlaps between our medium-sized ring library compounds and steroid natural products, especially with top-selling steroid drugs, suggesting the drug-likeness of our compounds. In addition, expanding the small rigid rings in steroid natural products to more flexible medium-sized rings covered broader chemical space occupied by steroid natural products and steroid drugs, implying the great potential of our compounds to occupy underexplored biological space. In two of the three variations (Fig. 6a and c), the medium-sized ring library occupied a distinct region of chemical space compared to the top-selling drugs. For example, our polycyclic compounds derived from steroid natural products generally have more stereogenic centers (average 7.4), OpreaRingCount (average 4.9), and rings (average 5.1), than those for drugs (average 2.7, 3.6, 3.7) (Supplementary Table 2).

We also carried out a principal moment of inertia (PMI) analysis to compare the three-dimensional shapes of the lowest-energy conformations of our library with the different reference sets as previously reported (Fig. 6d, for expanded version see Supplementary Fig. 4d)[53]. PMI analysis outlines differences of the three-dimensional molecular shapes of our medium-sized rings compared to the reference sets of libraries. Steroid drugs and diverse steroid and terpenoid natural products reside along the rod-disc (left) side

of the triangle, with a preference for the rod vertex. Although the distinctions between the different reference sets are less obvious in the PMI plot, we were pleased to find that our library covers a much broader three-dimensional chemical space than known steroid drugs and diverse steroid and terpenoid natural products.

## Discussion

In summary, we have developed a convenient approach for the rapid preparation of large numbers of polycyclic compounds. These small molecules boast diverse skeleta, multiple stereogenic centers, and medium-sized rings. Designing and synthesizing a library of this magnitude is no small feat, yet it is a timely addition to current compound libraries, most of which lack significant diversity. The success of our approach is predicated on the synergy that results from using state-of-the-art site-selective oxidation of C–H bonds in conjunction with ring expansion reactions. By transforming small, rigid rings into conformationally-labile medium-sized rings, this methodology can have significant biological implications. Medium-sized rings, which have multiple low energy conformations, can conceivably engage in multiple modes of binding, as opposed to small rings, which are often locked in a single conformation. Using chemoinformatics, we demonstrated that our synthesized compound library occupies unique chemical space compared to diverse reference compounds. This diversification strategy, while developed for steroid natural products, has the potential to be further expanded to other types of natural products or natural product-like compounds. We also demonstrated that our compounds with diverse scaffolds can be easily further derivatized, which is important for the discovery of bioactive small-molecule probes. In the future, natural product diversification by sequential C–H oxidation and ring expansion can be a general approach in the continued evolution of diversity-driven synthetic strategies. Our medium-sized

**Fig. 5** Further diversication of medium-sized ring scaffolds. **a** Formation of carbamates. **b** Formation of azide. **c**. Formation of imide **47**. **d** Formation of imide **48**. **e** Ring-cleavage of medium ring scaffold. A total of 150 polycyclic final products, most of which have >90% purity (LC-MS, ultraviolet detector at 254 or 210 nm) and >10 mg quantities, were prepared. (i) (a) CDI, Et$_3$N, CH$_2$Cl$_2$, rt, (b) R$^1$R$^2$NH, DMAP, toluene, 90 °C; (ii) NaH, HMPA, THF, then RI, rt; (iii) R$^1$R$^2$NH, toluene/CH$_2$Cl$_2$, rt, then 90 °C; (iv) (a) 5 M NaOH, THF, 60 °C, then NaBH$_4$, rt, then HCl, (b) RNH$_2$, toluene, rt, then 90 °C, (c) Dess-Martin oxidant, NaHCO$_3$, CH$_2$Cl$_2$; (v) (a) NaOH, EtOH/H$_2$O, 100 °C, (b) DMSO, HCl, 120 °C; (vi) (a) NaN$_3$, DMF, 80 °C, (b) Sodium ascorbate, CuSO$_5$•H$_2$O, HCCR, THF/H$_2$O, rt, (c) TsOH, THF/H$_2$O, rt; (vii) DPPA, Et$_3$N, R$^2$NH$_2$, DMF, rt. Full details are in the Supplementary Figures 13–16

ring library is now being screened against a wide range of challenging biological targets.

Breakthroughs in organic synthesis often bias the drug discovery landscape in the years that follow. As computational methods, automation, and high-throughput screening continue to gain momentum, unsolved problems in synthetic methodology can lead to drug development bottlenecks[54]. Our diversification strategy allows us to diversify natural products, easily synthesize medium-sized rings, a historical hurdle in organic synthesis, and access uncharted chemical space. We hope that the methods described herein will motivate others to look at organic synthesis as a call to action instead of a constraint.

## Methods

**General remarks**. All detailed synthetic methods are included in the supplementary information. Some representative general procedures are provided here.

**General procedure for the formation of β-keto ester**. A solution of diisopropylamine (1.3 equiv) in tetrahydrofuan (THF) was cooled to −78 °C. To the above solution was added n-BuLi (1.25 equiv) dropwise. The resulting solution was stirred for 45 min under argon. To the above reaction mixture was added a solution of ketone (1.0 equiv) in THF at −78 °C. The reaction mixture was kept for 2 h at this temperature. Ethyl cyanoformate (1.3 eq) was added to the above solution. The resulting mixture was stirred at −78 °C for 1 h. The reaction was allowed to warm

to room temperature. After quenching with NH$_4$Cl solution, the mixture was diluted with ethyl acetate and washed with brine (50 mL × 2). The combined organic phase was dried over sodium sulfate and concentrated under vacuum. Flash column chromatography over silica gel afforded the β-keto ester.

**General procedure for the alkylation of β-keto ester**. To a solution of β-keto ester (1.0 equiv) and hexamethylphosphoramide (HMPA) (2.0 equiv) in THF (10 mL/mmol) was added NaH (1.3 equiv) under argon. The reaction mixture was stirred for 1 h. Then alkyl iodide (5.0 equiv) was added. The mixture was stirred for 24 h. After quenching with NH$_4$Cl solution, the mixture was diluted with ethyl acetate and washed with NaS$_2$O$_3$ and brine. The organic phase was dried over sodium sulfate and concentrated under vacuum to afford the crude product. Flash column chromatography over silica gel afforded the product.

**General procedure of Backman rearrangement**. To a solution of ketone (1.0-equiv) in EtOH (10 mL/mmol) was added hydroxylamine hydrochloride (10.0 equiv) and KOAc (10.0 equiv). The reaction mixture was heated at reflux for 3 h. The mixture was cooled to room temperature. The volatile EtOH was removed under vacuum and the resulting residue was diluted with ethyl acetate and washed with brine (20 mL × 2). The organic phase was dried over sodium sulfate and concentrated under vacuum to afford crude product, which was directly used in the next step without further purification.

To a solution of the above crude product in pyridine (5.0 mL/mmol) was added p-toluenesulfonyl chloride (2.0 equiv) and dimethylaminopyridine (0.1 equiv). The resulting reaction mixture was heated at 60 °C overnight, cooled to room temperature, diluted with ethyl acetate (100 mL), and washed with 2 N HCl, NaHCO$_3$ and brine sequentially. The organic phase was dried over sodium sulfate

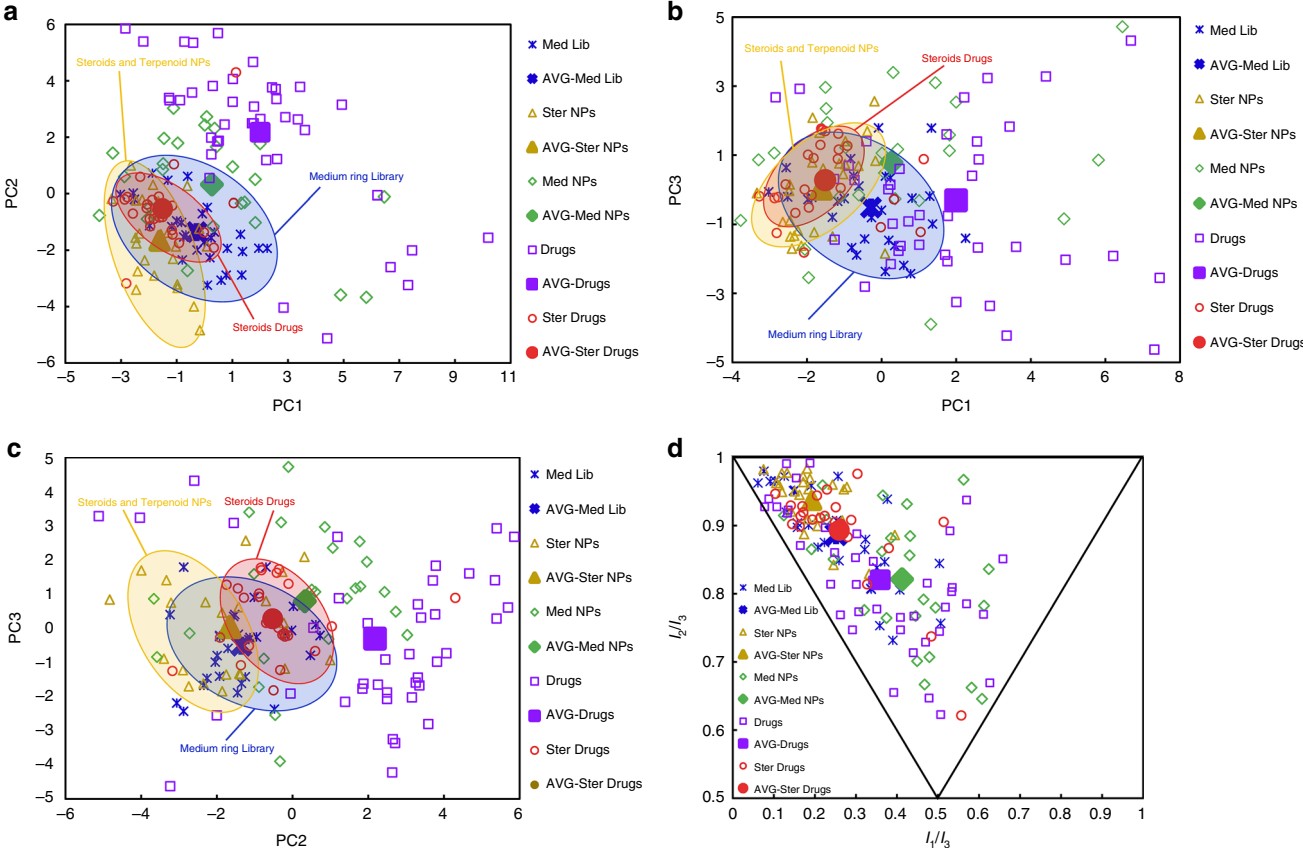

**Fig. 6** Cheminformatic analyses of polycyclic compounds with a medium-sized ring library. PCA and PMI plots of 30 medium-membered ring library (Med Lib) members, established reference sets of 40 brand-name drugs (without including the top-selling steroid drugs), 25 steroid drugs, 25 diverse steroids and terpenoid natural products and 25 diverse medium-membered ring natural products. The hypothetical average structure for each series (−AVG) is also shown. **a** PCA plot of PC1 versus PC2. **b** PCA plot of PC1 versus PC3. **c** PCA plot of PC2 versus PC3. The first three principal components account for 72% of the variance in the complete dataset, with individual contributions of 29.6%, 27.9%, and 14.4%, respectively. More than 90% of the variance in the complete 20-dimensional dataset is accounted for by the first six principal components (PC1–PC6). **d** PMI plot showing the three-dimensional shape of the lowest-energy conformations of each compound. Expanded PCA and PMI plots are in Supplementary Figures 4, and complete data are in Source Data

and concentrated under vacuum. Flash column chromatography over silica gel afforded the lactam product.

**General procedure for the formation of carbamates**. To a solution of alcohol (0.10 mol) in THF (2 mL) was added carbonyldiimidazole (32.4 mg, 0.20 mmol) and Et₃N (20.2 mg, 0.20 mmol). The reaction mixture was stirred at room temperature overnight. The volatile dichloromethane was removed under vacuum and the residue was dissolved in toluene (4 mL), amine (0.50 mmol), Et₃N (40.4 mg, 0.40 mmol), and dimethylaminopyridine (5.0 mg) were added to the reaction mixture. The reaction mixture was heated at 90 °C for 4 h. The mixture was cooled to room temperature then diluted with ethyl acetate (50 mL), and washed with brine (10 mL × 2). The organic phase was dried over sodium sulfate and concentrated under vacuum. Flash column chromatography over silica gel afforded the carbamates.

**Alternative procedure for the formation of carbamates**. To a solution of alcohol (0.10 mol) in dichloromethane (2 mL) was added *N*, *N*-dialkyl-1*H*-imidazole-1-carboxamide (0.20 mmol) and *t*-BuOK (22.4 mg, 0.20 mmol). The reaction mixture was stirred at room temperature overnight. The mixture was diluted with ethyl acetate (30 mL) and washed with brine (10 mL × 2). The organic phase was dried over sodium sulfate and concentrated under vacuum. Flash column chromatography over silica gel afforded the carbamates.

**General procedure for the formation of imides**. To a solution of **8** (44.2 mg, 0.1 mol) in dichloromethane (0.5 mL) and toluene (3.0 mL) was added amine (0.5 mmol). The reaction mixture was stirred for 30 min and then heated at reflux for 2 h. The mixture was cooled to room temperature, and concentrated under vacuum. Flash column chromatography over silica gel afforded the imide. (Note, DMF was used as solvent for amino acids).

**Alternative procedure for the formation of imides**. A solution of **6a** (0.50 mmol) in aqueous NaOH (30%, 5 mL) and THF (5 mL) was stirred for 2 h at 50 °C. The reaction mixture was cooled to 0 °C and then NaBH₄ (76 mg, 2.00 mmol) was added in one portion. The reaction mixture was allowed to warm to room temperature and stirred for 3 h. The reaction mixture was concentrated under vacuum, acidified by the slow addition of 1 N HCl, extracted with EtOAc (3 × 30 mL). The combined organic extracts were washed with H₂O (30 mL), brine (30 mL), and dried over Na₂SO₄. The organic phase was concentrated under vacuum and used for the next step.

To a solution of the above crude product in dichloromethane (0.5 mL) and toluene (3.0 ml) was added amine (0.5 mmol). The reaction mixture was stirred for 30 min then heated at reflux for 2 h. The mixture was cooled to room temperature, and concentrated under vacuum. Flash column chromatography over silica gel afforded the imide.

To a solution of the above imide in dichloromethane (5 mL) was added Dess-Martin periodinane (848 mg, 2.0 mmol). The reaction mixture was stirred at room temperature for 3 h. The mixture was diluted with ethyl acetate (30 mL) and washed with NaS₂O₃ (10 mL) and brine (10 mL × 2). The organic phase was dried over sodium sulfate and concentrated under vacuum. Flash column chromatography over silica gel afforded the final imide product.

## Data availability

For the details of the synthetic procedures, ¹H, ¹³C NMR and LC-MS spectra of the compounds in this manuscript, see Supplementary Methods. Data of chemoinformatic analysis are available in Source Data. The supplementary crystallographic data for this paper could be obtained free of charge from The Cambridge Crystallographic Data Centre (CCDC 1888660 and CCDC 1888661) via https://www.ccdc.cam.ac.uk/.

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

## Acknowledgements

We thank Prof. Dr. Shannon S. Stahl, Dr. Fei Wang and Dr. Mohammad Rafiee (UW-Madison) for helpful discussions of electrochemical allylic C–H oxidation. Support for this research was provided by the University of Wisconsin-Madison, Office of the Vice Chancellor for Research and Graduate Education with funding from the Wisconsin Alumni Research Foundation (WARF). This study made use of the Medicinal Chemistry Center at UW-Madison instrumentation, funded by the UW School of Pharmacy and WARF.

## Author contributions

C.Z. designed most of the scaffolds and conducted most of the experiments. Z.Y. and Z.-X. M. prepared a significant number of medium-sized ring compounds in the library. S.A.W. performed the cheminformatic analysis. S.A.B. made significant contributions to the writing and the formulation of some concepts. L.H. provided some of the natural product starting materials. I.A.G. analyzed all structures derived from X-ray crystallography. W.T. conceptualized and directed the project, and drafted the manuscript with the assistance from co-authors. All authors contributed to discussions.

## Additional information

**Competing interests:** The authors declare no competing interests.

