## [Peer Review File · Nature Communications]

Reviewers' comments:

Reviewer #1 (Remarks to the Author):

This paper by the Tang laboratory describes the synthesis of a library of steroid derivatives (along with a few other natural product types), in which medium size rings, both carboxylic and heterocyclic, have been incorporated using a variety of ring expansion reactions. This overall approach to library construction has been anticipated by published work by Hergenrother (ref 17) and, specifically in the steroid area, by Aube (ref 31). Conceptually, the most significant differentiation between this work and those precedents is the incorporation of Baran's synthetic strategy in which an "oxidation phase" is followed by a functional group manipulation phase (drawn in Figure 1 as "skeletal diversification" phase). This extends the scope of possible analogs to arise from (mostly) carbonyl groups that are not already present in the naturally occurring steroidal scaffolds used as starting structures. This attractively uses modern methods for oxidation, including electrochemistry, which is currently in vogue among synthetic organic chemists.

In considering whether to recommend this paper, the largest negative is that implied above: the novelty is modest given precedence. However, I think that this is offset by the overall quality of the organic chemistry depicted in the paper and the fact that the authors have made a whole lotta compounds, many of which have interesting structures. I tend to fall on the side of recommending acceptance pending a few items that require attention.

First, the introduction to this paper is painfully long and mostly dedicated to repeating the refrain that natural products are the most important compounds in medicinal chemistry. While I understand that many people think that this is so (for reasons ranging from protecting one's disciplinary turf to philosophical/religious arguments), I think it neither supported by facts (there are many synthetic compounds that have enriched humanity) nor necessary to build natural product-like structures up to be the sine qua non of library design. I do not wish to imply that natural products are not important, but rather to suggest that this rhetoric in the introduction could easily be toned down (while still supporting the present approach) en route to cutting the intro by about half. Similarly, the paragraph at the bottom of the first page disparaging combinatorial chemistry is simplistic at best and in a way ironic, since the present paper is essentially proposing a type of combinatorial chemistry.

Similarly, the author's claims of priority for medium size rings makes sense from a library enrichment perspective, but it is not true that they are missing from the pharmacopoeia (e.g., benzodiazepines, some tricyclic antidepressants, etc.) as suggested in the first paragraph.

A major concern that must be addressed prior to publication concerns the quantities of the compounds made and their purity. The authors only report yields in the experimental section (SI) but they should supply at least a range of quantities of the final library members prepared and details as to how purities were determined. I suspect that the response to the latter will be that the compounds looked pure by NMR (the spectra in the SI do in fact look OK), but it should be acknowledged that, for biological screening purposes, HPLC is the more commonly endorsed standard for purity. Also, near the end of the paper, the authors suggest that novel biological properties are expected to arise from these compounds. Since that is not described in the paper (and, rightly, beyond its scope) they should back off of such suggestions. One thing that they could do, beyond the PCA analyses which have become practically required in papers like this, would be to discuss some of the calculated physical properties of the various library members reported in the paper. Since they are likely to not lie within classical “drug-like” space, this point should at least be discussed.

Smaller items to consider in a revision:

1. The concept oxidation phase/functionalization phase summarized in Fig. 1 should be more explicitly attributed to Baran in the text.
2. The plural of “skeleton” is “skeleta” and not “skeletons”.
3. On page 3, the authors colloquially refer to an “[5] ring system”, but the more systematic nomenclature (“[6.3.0]”) would be better. In the same paragraph, they should define “DMAD”.
4. The IR listings in the SI are unnecessarily extensive; I do not suggest that they should be changed for this paper but in future papers the authors should adopt the modern rule that only important/assignable bands should be reported. Also, the units for IR bands are not wavelength (Greek nu) but wavenumbers (a Greek nu with a bar over it).
5. In Figure 3, the key oxidative transformations for parts c and d are not easy to see from the general structures used to introduce this scheme. Since this is where a larger part of the novelty of the paper lies, I recommend revising the scheme to clarify what the actual oxidation step does.
6. Page 5: What is an “Oprea ring count”? Reference, please.

Reviewer #2 (Remarks to the Author):

General comments:

The authors present a highly relevant study on a new strategy for the synthesis of natural product analogs with medium-sized rings. I can primarily comment on the cheminformatics analysis.

Whereas the overall science appears to be sound there seem to be some weaknesses primarily in reporting that need close attention, as I will point out below.

As a general comment, the authors should thoroughly check all citations in their work. Several citations seem to be not entirely accurate. For example, refs 5 and 8 are outdated; the reason for the citation of ref 38 is not understood; ref 39 seems to be a wrong reference, and several other references are incomplete or have typos.

Major comments:

Regarding the cheminformatics analysis, I have difficulties understanding the relevance of the reference datasets used in this work. Before this issue has been clarified I am unable to comment on the relevance of the conclusions drawn by the authors from this analysis.

- The set of 23 steroid drugs - where does this come from and what is the pharmaceutical relevance of these compounds (why have exactly those 23 compounds been picked and not others?)

- The set of 40 small-molecule drugs: The Author's citation suggests that they are coming from Ref 38, but reading this reference suggests that this is not the original source. From what I understand, these compounds are the 40 top-selling small-molecule drugs of 2006. Chemical space evolves quickly and this dataset appears to me as outdated. If my assumptions about the origin of this dataset are correct, the authors should redo this part of the analysis with an up-to-date and in particular, representative and unbiased reference dataset (e.g. including all FDA-approved drugs; note that all data can be included in the PCA but not necessarily all data points need to be presented in the graphs)

- The set of 36 steroid and terpenoid natural products with diverse structures: I don't understand where those come from and what their relevance is. Their relevance should be clearly stated.

- The 38 medium-sized ring natural products: Same as above - please assure that this actually is a representative set of structures. For example, include all medium-sized ring natural products from a large natural products dataset and motivate your choice.

I am unaware of MOE including descriptors for the number of atoms in aromatic rings, for the number of R/S centers, etc. Where do these descriptors come from? Are they coming from an SVL script/extension? If so, it should be clearly stated.

Several of the descriptors used, e.g. the number of h-bond acceptors, are dependent on the protonation state. Protonation states may also have an impact on the PMI analysis. However, as far as I understand, the protonation procedure (Wash function in MOE) is not specified. Please revise.

Referring to the previous comment, please ensure that the structures presented in the Supporting Information reflect those used for computing the descriptors, and add such a statement to the Supporting Information.

pg 6, "PMI analysis outlines differences of the three-dimensional... ..39". I don't understand why ref 39 is cited here. The statement refers to PMU analysis as a means to compare the molecular shape of compounds and the cited reference is a review of the importance of synthetic chemistry in the pharmaceutical industry.

Figure 6: Some of the key data points such as AVG-Stre Drugs in panel C are not visible. Please revise the figure accordingly.

Minor comments:

pg 1, "While natural products only represent 1% of published chemical structures..." Here and elsewhere in this paragraph, please state clearly that these are approximate numbers (e.g. "approximately 1%")

pg 1, "and antibacterial treatments.2-4". Here, also the latest review of Newman and Cragg should be cited (DOI: 10.1021/acs.jnatprod.5b01055)

pg 1, "At its outset in the 1990s..." Could the authors add a reference to support this statement?

pg 5, "The analysis included 40..." Please split this sentence for better readability.

pg 5, Same sentence: It is unclear why ref 38 is cited here - what it has to do with the statement.

Figure 6 caption and elsewhere, in particular also in the Supporting Information: "steroid drugs", not "Steroid drugs"

Figure 6 caption: "natural products.", not "natural product."

Ref 4: Please correct typo in the Author's name.

Ref 5 and 8 should be replaced with the latest review of Newman and Cragg (DOI 10.1021/acs.jnatprod.5b01055), and the statements in the manuscript text (in particular the one starting with "From the 1940s until 2010") updated accordingly.

Ref 21: Title missing.

Ref 38: Type on reference title.

Figure 6: The percentage of variance explained should be reported as part of the axis labels.

SI:

TOC: It seems that only Supplementary Figure 1 to 4 are listed in the TOC; the others are missing

Supplementary Figure 4: Why is this one included - isn't the content identical with that of Figure 6? Also, in Supplementary Figure 4a there is an issue with the transparency of the label "Steroids and Terpenoid NPs"

Something seems to have gone wrong with the page breaks. E.g., contrary to what is suggested in the caption of SI Figure 20, the figure is continued on the same page.

Supplementary Table 1 and respective Excel sheet: Some of the descriptor names are inaccurate. For example "Acceptor" should correctly be named "a_acc". Please check all names and revise as appropriate.

Comments from reviewer 1:

1.1) This paper by the Tang laboratory describes the synthesis of a library of steroid derivatives (along with a few other natural product types), in which medium size rings, both carboxylic and heterocyclic, have been incorporated using a variety of ring expansion reactions. This overall approach to library construction has been anticipated by published work by Hergenrother (ref 17) and, specifically in the steroid area, by Aube (ref 31). Conceptually, the most significant differentiation between this work and those precedents is the incorporation of Baran's synthetic strategy in which an "oxidation phase" is followed by a functional group manipulation phase (drawn in Figure 1 as "skeletal diversification" phase). This extends the scope of possible analogs to arise from (mostly) carbonyl groups that are not already present in the naturally occurring steroidal scaffolds used as starting structures. This attractively uses modern methods for oxidation, including electrochemistry, which is currently in vogue among synthetic organic chemists.

In considering whether to recommend this paper, the largest negative is that implied above: the novelty is modest given precedence. However, I think that this is offset by the overall quality of the organic chemistry depicted in the paper and the fact that the authors have made a whole lotta compounds, many of which have interesting structures. I tend to fall on the side of recommending acceptance pending a few items that require attention.

Our response:

We are grateful for the recommendation of acceptance of our manuscript by this reviewer. As pointed out by this reviewer, it is important to incorporate novel strategies and modern synthetic methods for the design of compound library to push the frontier of organic chemistry. We acknowledged that our strategy was inspired by Baran and others. We also want to point out the difference between our strategy and others.

As described in the revised manuscript, Baran's group elegantly completed the syntheses of several complex natural products by the two-phase strategy inspired by the biosynthetic pathways: 1) preparation of the diverse and complex hydrocarbon skeleta with a relatively lower oxidation stage in the "cyclization phase"; 2) selective C-H oxidation in the "oxidation phase". Inspired by Baran's two-phase strategy for the total synthesis of natural products, we proposed a two-phase strategy for the diversification of natural products: 1) selective C-H oxidation in the "Oxidation Diversity Phase"; 2) expansion of the small rings to medium-sized rings in the "Skeleton Diversity Phase". To the best of our knowledge, this type of two-phase diversification strategy has not been employed by anyone for the generation of compound libraries with diverse skeleta from natural products.

1.2) First, the introduction to this paper is painfully long and mostly dedicated to repeating the refrain that natural products are the most important compounds in medicinal chemistry. While I understand that many people think that this is so (for reasons ranging from protecting one's disciplinary turf to philosophical/religious

arguments), I think it neither supported by facts (there are many synthetic compounds that have enriched humanity) nor necessary to build natural product-like structures up to be the sine qua non of library design. I do not wish to imply that natural products are not important, but rather to suggest that this rhetoric in the introduction could easily be toned down (while still supporting the present approach) en route to cutting the intro by about half. Similarly, the paragraph at the bottom of the first page disparaging combinatorial chemistry is simplistic at best and in a way ironic, since the present paper is essentially proposing a type of combinatorial chemistry.

Our response:

We respect this reviewer's opinion and significantly shortened the introduction on the importance of natural products and removed the discussion about combinatorial chemistry. We condensed paragraphs 2-5 in the original manuscript to one paragraph in the revised manuscript.

1.3) Similarly, the author's claims of priority for medium size rings makes sense from a library enrichment perspective, but it is not true that they are missing from the pharmacopoeia (e.g., benzodiazepines, some tricyclic antidepressants, etc.) as suggested in the first paragraph.

Our response:

The sentence has been changed to "they were rarely found in the top 200 brand name and top 200 generic pharmaceuticals" in the revised manuscript.

1.4) A major concern that must be addressed prior to publication concerns the quantities of the compounds made and their purity. The authors only report yields in the experimental section (SI) but they should supply at least a range of quantities of the final library members prepared and details as to how purities were determined. I suspect that the response to the latter will be that the compounds looked pure by NMR (the spectra in the SI do in fact look OK), but it should be acknowledged that, for biological screening purposes, HPLC is the more commonly endorsed standard for purity.

Our response:

All of the final products were obtained over 10 mg and the amounts of each compound were added to the SI. To determine purity of our final library members, more than half of the compounds (77 compounds) were selected for LC-MS analysis. Among them, 74 compounds with UV absorbance showed >90% purity and the 3 compounds without UV absorbance showed >85% purity. Overall, 90% of the compounds showed >95% purity. These results including copies of the 77 LC-MS spectra were added to revised manuscript and SI. We believe the overall purity of the library is sufficient for future biological screening.

1.5) Also, near the end of the paper, the authors suggest that novel biological properties are expected to arise from these compounds. Since that is not described in the paper (and, rightly, beyond its scope) they should back off of such suggestions. One think that they could do, beyond the PCA analyses which have become

practically required in papers like this, would be to discuss some of the calculated physical properties of the various library members reported in the paper. Since they are likely to not lie within classical “drug-like” space, this point should at least be discussed.

Our response:

We are grateful for this comment and discussed this point in the manuscript. We added the following discussion to the manuscript.

“In all of the three variations, there are significant overlaps between our medium-sized ring library compounds and steroid natural products, especially with top-selling steroid drugs, suggesting the drug-likeness of our compounds. In addition, expanding the small rigid rings in steroid natural products to more flexible medium-sized rings covered broader chemical space occupied by steroid natural products and steroid drugs, implying the great potential of our compounds to occupy underexplored biological space. In two of the three variations, the medium-sized ring library occupy a distinct region of chemical space compared to the top-selling drugs (Fig. 6a, c). For example, our polycyclic compounds derived from steroid natural products generally have more stereogenic centers (average 7.4), OpreaRingCount (average 4.9), and rings (average 5.1), than those for drugs (average 2.7, 3.6, 3.7) (Supplementary Table 2).”

1.6) The concept oxidation phase/functionalization phase summarized in Fig. 1 should be more explicitly attributed to Baran in the text.

Our response:

Baran’s total synthesis strategy was inspired by nature’s two-phase biosynthesis pathways – cyclization phase and oxidation phase. We added the following to the revised manuscript.

“The chemistry underpinning their biosynthesis, particularly the oxidation component, inspired Baran’s group to complete several efficient total syntheses of complex natural products by preparing the hydrocarbon skeleta with low oxidation stage followed by selective C-H oxidation.

“Inspired by nature’s biosynthesis pathways and Baran’s two-phase strategy for the total synthesis of terpenes, we envisioned ...”.

1.7) The plural of “skeleton” is “skeleta” and not “skeletons”.

Our response:

All “skeletons” have been changed to “skeleta” in the revised manuscript.

1.8) On page 3, the authors colloquially refer to an “[5] ring system”, but the more systematic nomenclature (“[6.3.0]”) would be better. In the same paragraph, they should define “DMAD”.

Our response:

We have changed [7-5] to [5.3.0] and [8-5] to [6.3.0]. We also defined “DMAD” as dimethyl acetylenedicarboxylate.

1.9) The IR listings in the SI are unnecessarily extensive; I do not suggest that they should be changed for this paper but in future papers the authors should adopt the modern rule that only important/assignable bands should be reported. Also, the units for IR bands are not wavelength (Greek nu) but wavenumbers (a Greek nu with a bar over it).

Our response:

We thank this reviewer for the suggestion. We will follow the suggestion in the future. We also corrected the units for IR.

1.10) In Figure 3, the key oxidative transformations for parts c and d are not easy to see from the general structures used to introduce this scheme. Since this is where a larger part of the novelty of the paper lies, I recommend revising the scheme to clarify what the actual oxidation step does.

Our response:

We redrew figure 3 in the revised manuscript.

1.11) Page 5: What is an “Oprea ring count”? Reference, please.

Our response:

We added a reference (ref 52) in the revised manuscript.

Comments from reviewer 2:

2.1) As a general comment, the authors should thoroughly check all citations in their work. Several citations seem to be not entirely accurate. For example, refs 5 and 8 are outdated; the reason for the citation of ref 38 is not understood; ref 39 seems to be a wrong reference, and several other references are incomplete or have typos.

Our response:

We have checked all the references and corrected all the incomplete references and typos. We also reorganized the references according to format of Nature Communications.

2.2) Regarding the cheminformatics analysis, I have difficulties understanding the relevance of the reference datasets used in this work. Before this issue has been clarified I am unable to comment on the relevance of the conclusions drawn by the authors from this analysis.

- The set of 23 steroid drugs - where does this come from and what is the pharmaceutical relevance of these compounds (why have exactly those 23 compounds been picked and not others?)

Our response:

We reselected 25 steroid drugs as the reference. These steroid drugs are from the top selling and prescriptions small-molecule drugs in 2016 as described in the manuscript.

2.3) The set of 40 small-molecule drugs: The Author's citation suggests that they are coming from Ref 38, but reading this reference suggests that this is not the original source. From what I understand, these compounds are the 40 top-selling

small-molecule drugs of 2006. Chemical space evolves quickly and this dataset appears to me as outdated. If my assumptions about the origin of this dataset are correct, the authors should redo this part of the analysis with an up-to-date and in particular, representative and unbiased reference dataset (e.g. including all FDA-approved drugs; note that all data can be included in the PCA but not necessarily all data points need to be presented in the graphs)

Our response:

As described in the manuscript and SI, we have updated these 40 small-molecule drugs with top pharmaceutical products by retail sales in 2016 without including steroid drugs in this set.

2.4) The set of 36 steroid and terpenoid natural products with diverse structures: I don't understand where those come from and what their relevance is. Their relevance should be clearly stated.

Our response:

As described in the SI, we have reorganized the set of steroid and terpenoid natural products. In the updated set, we selected 25 steroid and terpenoid natural products as reference set. As indicated in the SI:

-The first seven compounds are starting materials we used for the preparation of our library.

-The next six compounds are steroids with different oxidative stage. These oxidative steroids were included because the increase of oxidative stage is the key diversity element in the first phase of our library synthesis.

-The remaining 12 terpenoid natural products represent skeleton diversity of steroid analogues because increase of skeleton diversity is the key element in the second phase of our library synthesis.

In summary, the set of 25 steroid and terpenoid natural products was selected to represent “oxidative stage diversity” and “skeleton diversity”.

2.5) The 38 medium-sized ring natural products: Same as above - please assure that this actually is a representative set of structures. For example, include all medium-sized ring natural products from a large natural products dataset and motivate your choice.

Our response:

As indicated in the SI, we have reorganized the set of medium-sized ring natural products and selected 25 of them as the reference set. These 25 selected compounds represent medium-sized ring natural products with 7-11 membered *N*-heterocyclic, *O*-heterocyclic and carbocyclic rings with skeleton and functional group diversity.

2.6) I am unaware of MOE including descriptors for the number of atoms in aromatic rings, for the number of R/S centers, etc. Where do these descriptors come from? Are they coming from an SVL script/extension? If so, it should be clearly stated.

Our response:

The MOE descriptor names are given in the table in SI. Descriptors that are not

explicitly part of the MOE Descriptor Calculation mode are all easily calculated from MOE SVL functions (aInRing, aInHRing, R and S from aRSChirality) or from combinations of other descriptor values (deltaRS, fChiralMW and fArRing).

2.7) Several of the descriptors used, e.g. the number of h-bond acceptors, are dependent on the protonation state. Protonation states may also have an impact on the PMI analysis. However, as far as I understand, the protonation procedure (Wash function in MOE) is not specified. Please revise.

Our response:

Compounds were loaded into MOE from SDF with explicit hydrogens in order to ensure correct chirality. The MOE Molecule Wash function was not used.

2.8) Referring to the previous comment, please ensure that the structures presented in the Supporting Information reflect those used for computing the descriptors, and add such a statement to the Supporting Information.

Our response:

We ensured that the structures presented in the SI reflect those used for computing the descriptors, and we have added such a statement to the revised SI.

2.9) pg 6, "PMI analysis outlines differences of the three-dimensional... ...39". I don't understand why ref 39 is cited here. The statement refers to PMU analysis as a means to compare the molecular shape of compounds and the cited reference is a review of the importance of synthetic chemistry in the pharmaceutical industry.

Our response:

This reference has been corrected.

2.10) Figure 6: Some of the key data points such as AVG-Stre Drugs in panel C are not visible. Please revise the figure accordingly.

Our response:

We redrew Figure 6.

2.11) pg 1, "While natural products only represent 1% of published chemical structures..." Here and elsewhere in this paragraph, please state clearly that these are approximate numbers (e.g. "approximately 1%")

Our response:

We shortened the discussion for the importance of natural product as suggested by reviewer 1. The above sentence was removed in the revised manuscript.

2.12) pg 1, "and antibacterial treatments.2-4". Here, also the latest review of Newman and Cragg should be cited (DOI: 10.1021/acs.jnatprod.5b01055)

Our response:

We added this reference.

2.13) pg 1, "At its outset in the 1990s..." Could the authors add a reference to support this statement?

Our response:

This paragraph has been deleted as suggestion by reviewer 1.

2.14) pg 5, "The analysis included 40..." Please split this sentence for better readability.

Our response:

The sentence has been replaced.

2.15) pg 5, Same sentence: It is unclear why ref 38 is cited here - what it has to do with the statement.

Figure 6 caption and elsewhere, in particular also in the Supporting Information: "steroid drugs", not "Steroid drugs"

Figure 6 caption: "natural products.", not "natural product."

Our response:

We deleted this reference. We also corrected "Steroid drugs" and "natural product" in the revised manuscript and SI.

2.16) Ref 4: Please correct typo in the Author's name.

Ref 5 and 8 should be replaced with the latest review of Newman and Cragg (DOI 10.1021/acs.jnatprod.5b01055), and the statements in the manuscript text (in particular the one starting with "From the 1940s until 2010") updated accordingly.

Ref 21: Title missing.

Ref 38: Type on reference title.

Our response:

We corrected all of the above references.

2.17) Figure 6: The percentage of variance explained should be reported as part of the axis labels.

Our response:

We added the percentage of variance in Figure 6.

2.18) TOC: It seems that only Supplementary Figure 1 to 4 are listed in the TOC; the others are missing.

Supplementary Figure 4: Why is this one included - isn't the content identical with that of Figure 6? Also, in Supplementary Figure 4a there is an issue with the transparency of the label "Steroids and Terpenoid NPs"

Our response:

We listed all the Supplementary Figures in TOC. Supplementary Figure 4 is the expanded version of Figure 6. We redrew Supplementary Figure 4a.

2.19) Something seems to have gone wrong with the page breaks. E.g., contrary to what is suggested in the caption of SI Figure 20, the figure is continued on the same page.

Our response:

We reorganized these figures in our revised SI.

2.20) Supplementary Table 1 and respective Excel sheet: Some of the descriptor names are inaccurate. For example, "Acceptor" should correctly be named "a_acc". Please check all names and revise as appropriate.

Our response:

We corrected all the descriptor names.

Reviewers' comments:

Reviewer #1 (Remarks to the Author):

This paper has been modestly reviewed by the authors, but all of my scientific concerns, notably those concerning establishment purity, have been addressed. The authors have also been responsive to my various questions pertaining to style and argument. I have no additional comments and support publications of this revised paper.

Reviewer #2 (Remarks to the Author):

The authors have addressed the majority of all issues raised. However, I am a bit surprised that apparently insufficient effort has been made to improve on some of the key issues that I have raised previously, which include in particular the accurate and proper referencing of literature (I am not referring to citation formats) and transparency about the origin of data. Both aspects are of critical importance to the reproducibility of the work.

In the following list I am naming four examples of remaining issues which are all taken from just a single paragraph of the manuscript.

The list should not be regarded as a complete list of open issues. I am willing to thoroughly review a further improved version of the manuscript.

1. Why do the authors cite ref 50 in the context of the statement "We used principal component analysis (PCA) to compare our compound library with approved therapeutics and natural products"? It is unclear what information is taken from ref 50 at this occasion. The citation seems to be a mistake.
2. "a collection of 40 top selling small-molecule drugs in 2016, ref 51". The authors cite reference 51 at this occasion, which reports on a small-molecule drugs data set from 2008, not 2016. Also, the collection published in reference 51 lists 200 drugs and not 40, for which reason it should be avoided to refer to a specific collection published in a specific article. It rather is a subset of the 40 top selling drugs compiled from [citation].
3. "a collection of 40 top selling small-molecule drugs in 2016, ref 51" Only from the Author's response to my previous questions I gather that these 40 compounds do not include steroid drugs. This is not explicitly mentioned in the main text as I believe. One could guess so because the authors include also a set of 25 of the top selling steroid drugs but this is not sufficiently accurate.
4. "25 steroid and terpenoid natural products with diverse oxidative stage and skeleta and 25 diverse natural products with a medium-sized ring": It is entirely unclear where those come from.

Additional comment: It is not clear what "variations" refers to in the statement "In all of the three variations..."

Comments from reviewer 1:

This paper has been modestly reviewed by the authors, but all of my scientific concerns, notably those concerning establishment purity, have been addressed. The authors have also been responsive to my various questions pertaining to style and argument. I have no additional comments and support publications of this revised paper.

Our response:

We are grateful that reviewer 1 supports the publication of the revised manuscript.

Comments from reviewer 2:

The authors have addressed the majority of all issues raised. However, I am a bit surprised that apparently insufficient effort has been made to improve on some of the key issues that I have raised previously, which include in particular the accurate and proper referencing of literature (I am not referring to citation formats) and transparency about the origin of data. Both aspects are of critical importance to the reproducibility of the work.

In the following list I am naming four examples of remaining issues which are all taken from just a single paragraph of the manuscript. The list should not be regarded as a complete list of open issues. I am willing to thoroughly review a further improved version of the manuscript.

1) Why do the authors cite ref 50 in the context of the statement "We used principal component analysis (PCA) to compare our compound library with approved therapeutics and natural products"? It is unclear what information is taken from ref 50 at this occasion. The citation seems to be a mistake.

2) "a collection of 40 top selling small-molecule drugs in 2016, ref 51". The authors cite reference 51 at this occasion, which reports on a small-molecule drugs data set from 2008, not 2016. Also, the collection published in reference 51 lists 200 drugs and not 40, for which reason it should be avoided to refer to a specific collection published in a specific article. It rather is a subset of the 40 top selling drugs compiled from [citation].

3) "a collection of 40 top selling small-molecule drugs in 2016, ref 51" Only from the Author's response to my previous questions I gather that these 40 compounds do not include steroid drugs. This is not explicitly mentioned in the main text as I believe. One could guess so because the authors include also a set of 25 of the top selling steroid drugs but this is not sufficiently accurate.

4) "25 steroid and terpenoid natural products with diverse oxidative stage and skeletal and 25 diverse natural products with a medium-sized ring": It is entirely unclear where those come from.

5) Additional comment: It is not clear what "variations" refers to in the statement "In all of the three variations..."

Our response:

We appreciate this reviewer's efforts on critically reading our revised manuscript and

providing additional suggestions to further improve the quality of our manuscript. We are grateful that reviewer 2 acknowledged our efforts on addressing the majority of issues raised. We apologize for overlooking some of issues related to the accurate and proper referencing of literature, particularly the previously cited reference 51.

1) We added a new reference (ref 49) on principal component analysis in the revised manuscript. We also added “following protocols employed by Tan (ref 50)” in the revised manuscript to clarify why ref 50 was cited.

2) We followed this reviewer’s suggestion and revised reference 51 as the following to clarify the situation: “The 40 top selling small-molecule drugs in 2016 and the 25 top selling small-molecule steroid drugs in 2016 are subsets of the 200 top selling drugs compiled and produced by Njardarson’s group. See details in <https://njardarson.lab.arizona.edu/content/top-pharmaceuticals-poster>. For the original reference, see: McGrath, N. A., Brichacek, M. & Njardarson, J. T. A graphical journey of innovative organic architectures that have improved our lives. *J. Chem. Educ.*, **87**, 1348–1349 (2010).”

3) We followed this review’s suggestion and added the statement of “(without including the top selling steroid drugs)” in the revised manuscript (both in the text and Figure 6) and SI.

4) In the revised manuscript, we added references “28, 29, 38” after the 25 steroid and terpenoid natural products, and added references “16, 25, 41, 50” after the 25 diverse natural products with a medium-sized ring. These natural products were selected from the above cited reviews. We also added the following statement to the revised manuscript: “The details of the selection criteria for the natural products are described in the Supplementary Information (page S93).”

We provided the following selection criteria for the natural products in page S93 of the SI. “The first seven compounds are starting materials we used for the preparation of our library. The next six compounds are steroids with different oxidative stage. These oxidative steroids were included because the increase of oxidative stage is the key diversity element in the first phase of our library synthesis. The remaining 12 terpenoid natural products represent skeleton diversity of steroid analogues because increase of skeleton diversity is the key element in the second phase of our library synthesis. In summary, the set of 25 steroid and terpenoid natural products was selected to represent “oxidative stage diversity” and “skeleton diversity”. The 25 diverse natural products with a medium-sized ring represent medium-sized ring natural products with 7-11 membered *N*-heterocyclic, *O*-heterocyclic and carbocyclic rings with skeleton and functional group diversity.”

5) In the revised manuscript, we changed “In all of the three variations, there are ...” to “As shown in Fig. 6a–c, there are...”. We also added “(Fig. 6a and 6c)” after “In two of the three variations...”.

Additional changes:

There are four paragraphs about the chemoinformatic analysis. In addition to the above issues raised by reviewer 2 for the first and third paragraphs, we also made the

following changes for the second and fourth paragraphs.

For the second paragraph:

We changed “As shown in Fig. 6, the parameters...” to “As shown in Fig. 6a-c, the parameters ...”

For the fourth paragraph:

We added “as previously reported (Fig. 6d)” and reference “53” after “We also carried out a principal moment of inertia (PMI) analysis to compare the three-dimensional shapes of the lowest-energy conformations of our library with the different reference sets”

In addition, we removed reference 10 as it is not closely related to the content. We then updated all relevant reference numbers.

I believe we have addressed all concerns raised by reviewer 2. Should you need any further information, please do not hesitate to contact me.

I look forward to hearing from you.

Sincerely,

Weiping Tang, PhD

Professor of Pharmaceutical Sciences and Organic Chemistry

School of Pharmacy and Department of Chemistry

University of Wisconsin-Madison

REVIEWERS' COMMENTS:

Reviewer #2 (Remarks to the Author):

All my comments have been adequately addressed by the authors. I have no further requests and recommend acceptance of the manuscript for publication.